# Perspectives for Glyco-Engineering of Recombinant Biopharmaceuticals from Microalgae

**DOI:** 10.3390/cells9030633

**Published:** 2020-03-05

**Authors:** Lorenzo Barolo, Raffaela M. Abbriano, Audrey S. Commault, Jestin George, Tim Kahlke, Michele Fabris, Matthew P. Padula, Angelo Lopez, Peter J. Ralph, Mathieu Pernice

**Affiliations:** 1Climate Change Cluster, University of Technology Sydney, Broadway Campus, Ultimo NSW 2007, Sydney, Australia; Raffaela.AbbrianoBurke@uts.edu.au (R.M.A.); Audrey.Commault@uts.edu.au (A.S.C.); Jestin.George@student.uts.edu.au (J.G.); Tim.Kahlke@uts.edu.au (T.K.); Michele.Fabris@uts.edu.au (M.F.); Peter.Ralph@uts.edu.au (P.J.R.); 2CSIRO Synthetic Biology Future Science Platform, Brisbane, QLD 4001, Australia; 3School of Life Sciences and Proteomics Core Facility, Faculty of Science, University of Technology Sydney, Ultimo NSW 2007, Sydney, Australia; Matthew.Padula@uts.edu.au; 4Department of Chemistry, University of York, York, YO10 5DD, UK; al1510@york.ac.uk

**Keywords:** algae, glycosylation, recombinant protein

## Abstract

Microalgae exhibit great potential for recombinant therapeutic protein production, due to lower production costs, immunity to human pathogens, and advanced genetic toolkits. However, a fundamental aspect to consider for recombinant biopharmaceutical production is the presence of correct post-translational modifications. Multiple recent studies focusing on glycosylation in microalgae have revealed unique species-specific patterns absent in humans. Glycosylation is particularly important for protein function and is directly responsible for recombinant biopharmaceutical immunogenicity. Therefore, it is necessary to fully characterise this key feature in microalgae before these organisms can be established as industrially relevant microbial biofactories. Here, we review the work done to date on production of recombinant biopharmaceuticals in microalgae, experimental and computational evidence for *N*- and *O*-glycosylation in diverse microalgal groups, established approaches for glyco-engineering, and perspectives for their application in microalgal systems. The insights from this review may be applied to future glyco-engineering attempts to humanize recombinant therapeutic proteins and to potentially obtain cheaper, fully functional biopharmaceuticals from microalgae.

## 1. Introduction

Biopharmaceuticals are biological macromolecules that exhibit therapeutic actions in humans. This group of compounds includes essential molecules such as antibodies, hormones, and vaccines [1]. More than 60% of commercialized biopharmaceuticals are recombinant proteins [2], which are produced in genetically engineered host cells defined as biofactories [3]. Biofactories can include a broad spectrum of host organisms spanning from prokaryotic to eukaryotic cells and support a US$140 billion biopharmaceuticals market [4]. 

Currently, the host cell systems *Escherichia coli* (bacteria), *Saccharomyces cerevisiae* and *Pichia pastoris* (yeast), and Chinese hamster ovary (CHO) cells (mammalian) dominate the production of biopharmaceuticals, with the mammalian CHO cells covering >50% of the market [5]. However, these four cell systems have several disadvantages, including species-specific issues related to the nature of each biofactory. *E. coli* often incurs translational errors, accumulates inclusion bodies, and completely lacks the eukaryotic organelles and machinery necessary to produce fundamental post-translational modifications (PTMs) [6]. Challenges with using yeasts include inadequate protein secretion and incorrect protein PTMs [7,8]. Biopharmaceutical production with CHO cells is expensive because of the complex culturing requirements associated, difficult to scale, and susceptible to contamination with human viruses and prions [9,10]. These complications have prompted efforts to optimize these systems as well as identify more suitable host cell lines. In this context, microalgae have emerged as attractive novel expression systems for biopharmaceutical production.

*Chlamydomonas reinhardtii* (Chlorophyceae) has historically been the model microalgal species for biotechnological innovations [11], due to its advantageous biological features [12,13]. However, other species are increasingly being evaluated for their capacity to produce recombinant proteins. Microalgae including *Chlorella sp.* (Trebouxiophyceae) [14,15], *Dunaliella salina* (Chlorophyceae) [16], *Nannochloropsis oculata* (Eustigmatophyceae) [17,18] and the diatom *Phaeodactylum tricornutum* (Bacillariophyceae) [19] have been reported to successfully express biopharmaceuticals.

A major challenge with producing biopharmaceuticals in non-human cells is obtaining correct post-translational modifications (PTMs) of the recombinant protein. Since each host cell system possesses its own unique protein processing machinery at a translational and post-translational level, the same recombinant protein produced in different cell systems can display different and unique PTMs [20,21,22,23,24]. PTMs such as phosphorylation, acetylation, methionine oxidation, asparagine and glutamine deamination, proteolysis, disulphide bond formation, and glycosylation all play a significant role in stability, functionality, and activity of proteins [25]. Among these, glycosylation is a major PTM that is found on more than 50% of human proteins [26]. In biopharmaceutical production, over 40% of approved therapeutic proteins are glycosylated, highlighting the importance of understanding and controlling the glycosylation mechanism of non-human expression systems [27,28].

The cellular mechanisms that determine protein glycosylation patterns are complex and vary among different eukaryotic species [29]. Given the significant influence of glycosylation on yield, efficacy, pharmacokinetics, and immunogenicity of recombinant therapeutic proteins, it is essential to choose the right host expression system to successfully produce a functional biopharmaceutical [30]. Additionally, sub-optimal glycosylation of recombinant proteins can be overcome using glyco-engineering strategies [31]. Glyco-engineered biopharmaceuticals will present “humanized” glycans that will not be immunogenic to humans [4].

In this review, we provide a brief overview of various host systems, with a focus on the advantages and disadvantages of microalgae as biofactories for the production of recombinant therapeutic proteins. We also discuss strategies to overcome the challenges, with a focus on microalgae glycosylation status and its comparison to human glycosylation. Lastly, we assess the prospect of applying glyco-engineering techniques to optimize recombinant biopharmaceutical production in microalgal host systems.

## 2. Production of Biopharmaceuticals in Alternative Hosts

Due to improvements in genetic engineering technologies and recombinant protein expression, the repertoire of biopharmaceuticals produced in biofactories is expanding. Increased capability to replicate the functional characteristics of larger, more complex proteins [32] has broadened the range of potential biopharmaceutical applications, including treatments for cancer and autoimmune diseases [33]. This is particularly evident for antibodies [32], which have been replicated in various forms including single-chain fragment variables (scFvs) [34] and antigen-binding fragments (Fabs) [35], amongst others. The choice of biofactory is primarily based on the type of biopharmaceutical (e.g., functional multi-domain mammalian proteins cannot be produced in bacteria as they require specific PTMs), but other factors such as production costs, yields, time to market, and the safety of the patient are also considered when selecting an expression host. The recent production of a biopharmaceutical used as a treatment for the Ebola virus in Western Africa has proven that plants (in this case *Nicotiana benthamiana*) can be a powerful alternative to common expression systems, especially for rapid and low-cost production of antibodies using technologies that are relatively simple to implement in developing countries [36,37,38]. Among the multitude of alternative expression hosts, photosynthetic systems have emerged with high potential to offer a rapid and cost-effective alternative to traditional hosts. The previously described traditional biofactories are all heterotrophic systems and rely on the addition of organic carbon sources to the culture media, whilst photosynthetic organisms can generate biomass using CO_2_ and light, lowering not only carbon footprint but also media complexity and costs associated with recombinant protein production. In recent decades, plant cells and microalgal organisms have been analysed for their promising role as photosynthetic host systems, with the latter being the focus of this review.

## 3. Microalgal Biofactories

Microalgae are unicellular photosynthetic organisms found in both terrestrial and aquatic environments [39]. The potential of microalgae to produce natural products with commercial value, ranging from food additives and health supplements to biofuels and cosmetics, has currently become widely recognized [40]. Recent advances in genetic engineering have enabled expression of many recombinant biopharmaceuticals in microalgal hosts (Table 1).

Microalgal biofactories possess many unique advantages. Like plants, microalgae can be cultivated in large areas and require a lower up-front investment compared to bacterial or mammalian cell systems [41]. Additionally, microalgae do not harbor human pathogens and some species are generally recognized as safe (GRAS) [12]. They are easy to cultivate in bioreactors, and thus are less prone to airborne contamination issues [12]. Moreover, microalgae have higher growth rates and less complex media requirements compared to plants, and therefore potentially lower production costs [10,11,12]. Furthermore, microalgae are unicellular organisms that lack functional parts such as roots and leaves, making recombinant biopharmaceuticals production more uniform within the same batch and diminishing the energy and resources needed to generate additional biomass [12].

One of the better characterized and widely exploited microalgae for recombinant protein production is the biflagellate chlorophyte *C. reinhardtii* [42]. This species has become the model algae to explore and understand the biological processes occurring within the green microalgal lineage due to an advanced genetic toolkit and the availability of fully sequenced nuclear and organellar genomes [12]. *C. reinhardtii* is now the most comprehensive microalgal platform for expression of recombinant proteins with promising industry applications in the bioenergy, biopharmaceutical, biomaterial, and nutraceutical sectors [11,43]. A recent toxicology study demonstrated that *C. reinhardtii* biomass is safe for human consumption [44], and although *C. reinhardtii*-derived products are not yet commercially available, extensive research on large-scale cultivation is bringing it closer to reality [45].

Recombinant protein production in *C. reinhardtii* has been achieved by engineering both the chloroplast and nuclear genomes. In the chloroplast, the level of transgenic expression can reach 20% of total soluble protein (TSP) [10]. However, recombinant proteins expressed in the chloroplast are retained inside the plastid and cannot be secreted [46]. Therefore, recombinant proteins expressed in the chloroplast are unable to pass through the secretion route via the endoplasmic reticulum and the Golgi apparatus, and cannot be subjected to fundamental PTMs such as glycosylation [12,46]. Hence, chloroplast expression is only suitable for non-glycosylated proteins. Recombinant proteins expressed from the nucleus, on the other hand, can be targeted to the secretion route by adding specific signal peptides to the recombinant amino acid sequence, resulting in secretion and glycosylation of the recombinant protein [10]. Unfortunately, nuclear expression results in very low yields, due to random integration, low transformation efficiency, and gene silencing mechanisms [46,47].

As increasing knowledge and genetic tools become available, other microalgae species have been exploited for recombinant protein production, including monoclonal antibodies, hormones, and enzymes. Several species, including *Chlorella vulgaris, Chlorella sorokiniana*, *Chlorella ellipsoidea*, *D. salina, N. oculata*, and the diatom *P. tricornutum* have been successfully used to express biopharmaceuticals (Table 1). In particular, *P. tricornutum* has gained noticeable importance due to the ability to secrete fully functional IgG antibodies [19]. An overview of the recombinant biopharmaceuticals produced in microalgae over the last two decades is presented in Table 1.

Few recombinant biopharmaceuticals produced in alternative host systems have successfully passed clinical trials and reached commercialization status [2,78]. One is ELELYSO^®^, a recombinant glucocerebrosidase produced in carrot cells and approved for Gaucher’s disease treatment by FDA in 2012 [2,79]. The second one is ZMapp, an antibody cocktail administered during the Ebola virus outbreak in Western Africa [36,37,38]. One component of ZMapp (cZMAb) was produced in *N. benthamiana* [37,38]. Recently, two new antibody therapies outperformed ZMapp, leading to a different approved therapy for the Ebola virus [80]. Microalgal biopharmaceuticals are still absent from the market, mainly due to the low yields of recombinant proteins obtained from these biofactories [46,47]. 

## 4. Post-Translational Modifications and Glycosylation

Post-translational modifications (PTMs) are chemical modifications of a protein during or after its synthesis within the cell. Amino acids, the building blocks that define the physio-chemical structure and functionality of a protein, can be altered by more than 250 different PTMs [81]. The repertoire of PTMs is very complex, considering that 15 out of the 20 common amino acids can be modified [82]. More than 5% of the total human genome encodes for enzymes involved in PTMs, including those involved in phosphorylation (kinases and phosphatases), acetylation (acetylases and deacetylases), and glycosylation (glycosyltransferases) [27]. Protein PTMs, together with alternative RNA splicing and translation, enhance molecular diversification of gene products and participate in a complex system to regulate the physiology of eukaryotic cells [27,83]. 

PTMs such as phosphorylation, glycosylation, and nitration are involved in important cellular processes [84]. For example, phosphorylation can operate as switches to modulate specific catalytic activities of the protein [85]. Another function of PTMs is to mark proteins for degradation by ubiquitination [85]. PTMs can also act in response to external stimuli, for example, when a cell is subjected to biological stress it can activate proteins with specific PTMs to counteract the stress [86]. Of the many PTMs relevant to biopharmaceutical production, glycosylation plays a major role. In fact, glycosylation is found on more than 50% of human proteins [26], and on more than 40% of approved recombinant biopharmaceuticals, highlighting the importance to understand and control the glycosylation mechanism of non-human expression systems [27,28].

Glycosylation refers to a covalent bond between a polysaccharide chain and an amino acid, formed during translation of the protein. The two most frequent types of glycosylation are *N*-linked and *O*-linked glycosylation. *N*-linked glycosylation is characterized by the formation of a covalent bond between the glycan and the amidic group of an asparagine (Asn) residue. *O*-linked glycosylated proteins have the glycan linked to the hydroxyl component of a serine (Ser) or a threonine (Thr) residue. Glycosylated proteins and glycan structures strongly regulate fundamental biological processes within the cell, such as cell adhesion, self/nonself recognition, molecular trafficking and clearance, receptor activation, and endocytosis [87].

Glycosylation significantly enhances yield, folding, efficacy, and pharmacokinetics of recombinant biopharmaceuticals [30]. During recombinant protein production, non-human host organisms can attach glycan residues (monosaccharides) that would be absent on the human endogenous protein, potentially resulting in lower yields, incorrect folding, and inefficacy of the biopharmaceutical [10]. For example, glycosylation plays a fundamental role in activity of antibody-based therapeutics [88]. In fact, antibodies can target and kill hostile cells by antibody-dependent cellular cytotoxicity (ADCC) or complement-dependent cytotoxicity (CDC) mechanisms, and both ADCC and CDC are directly related to glycosylation presence and status of the antibody [89,90,91]. Therefore, an antibody carrying incorrect glycosylation might show diminished activity. Moreover, correct glycosylation and proper folding also affect immunogenicity and antigenicity of recombinant biopharmaceuticals. Immunogenicity is the capability of a molecule to trigger an immune response in the patient, whilst antigenicity is the ability of a molecule to bind immune system products. Therefore, an antigen is not necessarily an immunogen, whilst an immunogen is inevitably an antigen. Immunogenic biopharmaceuticals can trigger an immune response in the patient resulting in accelerating clearance during therapy or, in some rare cases, life threatening complications [92]. Glycans can trigger an immunogenic reaction either indirectly or directly. Glycosylation can have an indirect effect on immunogenicity by influencing therapeutic proteins folding, solubility and structural stability [93]. In fact, incorrect or no glycosylation can alter the secondary/tertiary structure and/or prompt aggregation of therapeutic proteins, factors breaking the immune tolerance of the patient. Moreover, specific non-human residues can be directly recognised as exogenous by the patient immune system and trigger an immunogenic response [94]. At least four non-human glycans have been identified as being able to induce an immune response in humans. These residues are: α-Gal, Neu5Gc, β(1,2)-xylose and α(1,3)-fucose [93]. The α-Gal and Neu5Gc residues are present in therapeutics produced in mammalian cells such as CHO cells, while β(1,2)-xylose and α(1,3)-fucose are present in plant and microalgal-produced glycoproteins [93]. Given the crucial role of glycosylation in folding, activity, and immunogenicity of biopharmaceuticals, it is essential to understand the glycosylation capabilities of a chosen biofactory to produce properly folded, effective, and safe recombinant therapeutics.

### 4.1. N-Glycosylation

*N*-glycosylation requires a strict consensus sequence, where the amino acid sequence must be Asn-Xxx-Ser/Thr [95], where Xxx can be any amino acid other than proline (Pro). Further studies expanded this sequence to a less frequent, but still relevant, Asn-Xxx-Cysteine (Cys), Asn-Xxx-Valine (Val), and Asn-Glycine (Gly) [96].

*N*-glycosylation begins in the endoplasmic reticulum (ER), where a biosynthetic precursor, a dolichol-P-P-linked oligosaccharide (comprised of 3 glucose (Glu), 9 mannose (Man) and 2 *N*-acetylglucosamine (GlcNAc) residues) is transferred to Asn residues in nascent polypeptide chains (Figure 1). The glycan is then subjected to further enzymatic maturation as part of a quality control by chaperones (calnexin and calreticulin). The newly formed glycoprotein is then transferred to the Golgi apparatus and further modified by many different glycosyltransferases, until reaching final maturation. *N*-glycans across different eukaryotic organisms present a common structure called the “pentasaccharide core” (2 *N*-acetylglucosamine (GlcNAc) and 3 mannose (Man) residues) [10,97] (Figure 1). The rest of the residues will attach onto the final two Man of the pentasaccharide core, thus creating two polysaccharide antennae. However, unlike the well-conserved pentasaccharide core, the rest of the *N*-glycan structure vastly varies amongst eukaryotes. In fact, the final maturation of the glycan in the Golgi apparatus by different glycosyltransferases is a species-specific mechanism and is the source of different glycosylation patterns among eukaryotic organisms [98,99] (Figure 1).

In humans, depending on how the two terminal Man residues are elongated, three different specific *N*-glycans structures are possible: (i) “oligo-mannose” glycans containing only mannose residues, (ii) “hybrid” glycans with mannose residues on one antenna and mixed monosaccharides on the second antenna, and (iii) “complex” glycans with mixed monosaccharides on both antennae [100]. As a glycoprotein enters the Golgi, it is *N*-linked to 8/9 Man and 2 GlcNAc. This chain is reduced to 5 Man and 2 GlcNAc by an enzyme called α-mannosidase I (α-Man I) (Figure 1). Then, the *N*-acetylglucosaminyltransferase I (GnT I) plays a key role by transferring one *N*-acetylglucosamine residue on the α(1,3)-mannose arm of the 5 Man and 2 GlcNAc *N*-linked protein. This structure (1 GlcNAc, 5 Man and 2 GlcNAc) is the starting point for both “hybrid” and “complex” structures (Figure 1). From that point on, many different glycosyltransferases will build the complete pattern of the *N*-glycoprotein [101].

Bacteria, yeast, CHO, and plant glycoproteins possess species-specific *N*-glycans that differ from human *N*-glycans (Figure 1B). Glycosylation and PTMs in general are challenging issues for *E. coli*, as most prokaryotes lack the eukaryotic PTM machinery to perform glycosylation [6]. It is possible to transfer the basic prokaryotic glycosylation machinery of another bacteria (*Campylobacter jejuni*) into *E. coli* to obtain glycosylated recombinant biopharmaceuticals [102]. However, prokaryotic glycans do not show similarities with human glycosylation patterns, resulting in immunogenic biopharmaceuticals [102,103]. Yeast glycans present an excess of mannose residues assembled together in “hyper-mannosidic” structures [104], which greatly differ from the human patterns [4,105]. Although CHO cells possess human-like glycosylation machinery, some discrepancies still persist [4]. For example, the absence of fundamental human residues like α(2,6)-sialic acid and α(1,4)-fucose, and the production of undesired non-human residues such as *N*-glycolylneuraminic acid (Neu5Gc) and galactose-α(1,3)-galactose (α-Gal), can result in the production of potentially immunogenic recombinant biopharmaceuticals [4,33]. Similarly, plant cells produce glycans containing immunogenic residues such as β(1,2)-xylose and core α(1,3)-fucose [106]. 

Compared to other systems, very little is known about *N*-glycosylation in microalgae. The combination of genomic annotation and experimental evidence has revealed some details about *N*-glycans in some species; however, more information is needed. Analysis of five different microalgal species [94,107,108,109,110,111] showed two different glycosylation pathways based on presence or absence of the GnT I enzyme. The green microalgae *C. reinhardtii* and *C. vulgaris*, and the red microalga *Porphyridium purpureum* lack GnT I, which has been validated experimentally [94,107,108,111]. Thus, the *N*-glycosylation pathway in these species is defined as GnT I-independent. In this pathway, the 5 Man and 2 GlcNAc *N*-linked protein is subjected to the action of xylosyltransferases (XyT) and methyltransferases (MeT), leading to unique *N*-linked structures containing methylated mannoses linked to one or two xyloses (Figure 1). The structures vary slightly among these microalgae, with different possible locations of the xylose residues [94,107,108,111]. On the other hand, Baïet and colleagues [109] demonstrated that GnT I is present and active in the diatom *P. tricornutum*. In this case, GnT I transfers an *N*-acetylglucosamine residue to the 5 Man and 2 GlcNAc *N*-linked protein in the Golgi apparatus. The structure is then subjected to α-mannosidase II (α-Man II) and fucosyltransferase (FuT), resulting in paucimannosidic (Man_3–4_GlcNAc_2_) fucosylated *N*-glycans [109] (Figure 1C). The GnT I-dependent pathway is also present in the green microalga *Botryococcus braunii* [110]. *N*-linked glycans in *B. braunii* present methylation of mannose residues (absent in *P. tricornutum*) and the terminal GlcNAc (linked to the α(1,3)-mannose arm) can be attached to an additional hexose [110] (Figure 1C). 

Due to the presence of a eukaryotic PTM machinery and several different glycosidases, both GnT I-independent and GnT I-dependent microalgal species show *N*-glycosylation patterns more similar to humans than *E. coli* (glycosylation absent) and yeasts (“hyper-mannosidic” *N*-glycans). Nevertheless, discrepancies between human and microalgal *N*-glycans are still relevant. The absence of GnT I enzyme in GnT I-independent species is a significant issue. GnT I activity serves as starting point to produce “complex” and “hybrid” glycans, and its absence prohibits the construction of two out of three possible human *N*-glycan structures. Moreover, both GnT I-independent species listed here present abundant methylation of residues (absent in humans) and *C. reinhardtii* also presents attachment of Xyl residues, which is another immunogenic trait. Unsurprisingly, GnT I-dependent microalgae resemble human glycosylation more than GnT I-independent species. *B. braunii* shows native *N*-glycans similar to human “hybrid” structures, and *P. tricornutum* presents paucimannosidic glycans, an important pattern found in a specific class of biopharmaceuticals (as explained in Section 5.1.2) [79,112]. Nonetheless, α(1,3)-fucose residues in *P. tricornutum* [113] and methylation of mannoses in *B. braunii* [110] are both *N*-glycan characteristics absent in humans.

### 4.2. O-Glycosylation

*O*-glycosylation involves an oxygen-carbon bond between the hydroxyl group of a Ser or Thr residue of the protein and the polysaccharide chain (Figure 2). There are 7 subclasses of *O*-glycans, based on which monosaccharide is directly attached to the protein [114]. In humans, the most frequently observed *O*-glycoproteins are mucins and proteoglycans [106,114] (Figure 2). The first monosaccharide attached to a mucin protein is an *N*-acetylgalactosamine (GalNAc), usually followed by galactose (Gal) or a GlcNAc [106]. On the other hand, xylose is the first monosaccharide attached to a proteoglycan, followed by Gal [115]. Other important *O*-linked monosaccharides that can initiate the polysaccharide chain are Fuc and Man; *O*-fucosylation plays a fundamental role in transmembrane signalling [116,117] and *O*-mannosylation is involved in muscle and brain development [118]. While the *O*-GalNAcylation starts directly in the Golgi apparatus, all the other structures begin in the ER and the whole polysaccharide is later synthesized in the Golgi, like for *N*-glycans synthesis [114].

In most organisms including humans, *O*-glycosylation does not present a common structure or a consensus sequence, and many different structures are possible. Nevertheless, unique species-specific *O*-glycans are still recognisable in some organisms. *E. coli* lacks the eukaryotic organelles and PTM machinery to perform *O*-glycosylation [6]. *O*-glycans in yeasts present a core structure composed of a Man residue attached to a Ser or a Thr [119]. This structure presents a core Man with multiple mannoses attached, resulting in yeast-specific high-mannose *O*-glycans [119] (Figure 2). The majority of *O*-glycans in CHO cells present a core structure with a GalNAc residue attached to a Ser or a Thr, the same core structure as human mucins [120]. The GalNAc residue can be further linked to a Gal or a GlcNAc residue [120,121] (Figure 2). *O*-glycosylation core structures in plants can present a Gal residue attached to a Ser or a unique arabinose (Ara) residue attached to a hydroxyproline (Hyp) amino acid [122]. Recombinant Interferon (IFN) alpha 2b expressed in tobacco cells showed a unique *O*-glycan pattern with a Hyp-*O*-Gal core and several Ara and Gal residues [123]. 

To date, there is only one study reporting *O*-glycosylation analysis of proteins produced in microalgae. Bollig and colleagues [124] analysed linear hydroxyproline-bound *O*-glycans native proteins of the green alga *C. reinhardtii*, showing similarities and differences with higher plant *O*-glycans. They found the same *O*-glycoprotein core as in plants (Hyp-*O*-Ara-Ara), suggesting conservation within the green lineage. However, they also found a higher heterogeneity of glycans in *C. reinhardtii*, with the presence of galactofuran residues and methylated residues (absent in plants) [124]. Based on the structures experimentally characterized, they speculated about the *O*-glycosylation pathways in *C. reinhardtii*. They proposed that two arabinosyltransferases add the first two Ara residues to Hyp, and that a galactofuranosyltransferase and two methyltransferases specific to *C. reinhardtii* perform the final modifications [124].

While there is minimal experimental information available on *O*-glycosylation in microalgae, existing data suggests key differences with human glycosylation patterns. *C. reinhardtii* native *O*-glycoproteins present methylated residues on *O*-glycans, a trait also found in microalgal *N*-glycans but completely absent in humans. Moreover, the Hyp-*O*-Ara core has not been found in humans, suggesting the possibility of immunogenic activity for these types of *O*-glycans. However, further investigation of microalgal *O*-glycosylation is unquestionably needed; for example, it is not known to what extent the *C. reinhardtii O*-glycan core structure is immunogenic, if other species present similar *O*-glycan core structures, and whether these glycosylation patterns will also be found on recombinant proteins. In conclusion, more research must be conducted to unravel the *O*-glycosylation patterns in microalgae.

### 4.3. Computationally Predicted Distribution of Microalgal N- and O- Glycosylation Enzymes 

Experimental evidence from a limited number of microalgal species hints at a wide diversity of *N*-glycosylation patterns among different microalgal taxa. For example, the major glycosylation differences in two species (*C. reinhardtii* and *B. braunii*) belonging to the same phylum (Chlorophyta) suggest an extensive level of variation. On the other hand, it is difficult to assess the diversity of *O*-glycoproteins in microalgae, as *O*-glycosylation has not been adequately investigated across multiple groups. Although information from experimental characterization remains limited, computational analysis of available algal genomes permits the hypothetical reconstruction of protein *N*- and *O*- glycosylation pathways (Figure 3).

The genome assemblies of genomes *C. reinhardtii* (GCA_000002595), *P. purpureum* (GCA_000397085), *P. tricornutum* (GCF_000150955)*, B. braunii* (GCA_002005505), *C. vulgaris* (GCA_001021125), and *N. gaditana* (GCA_000569095; GCF_000240725; GCA_001614215), were downloaded from the National Center for Biotechnology Information (NCBI). Potential Open Reading Frames (ORF) sequences were predicted on all assemblies using EMBOSS’ *getorf* command [125]. Gene candidates were identified using the Basic Local Alignment Tool (BLAST) [126] searching all predicted ORFs for a list of candidate genes. Additionally, template genes were searched for functional domains using PFAM [127] and HMMER v3 [128]. Hit-domain motifs were downloaded and search for in the predicted ORFs. All candidate ORFs determined with BLAST and Pfam domain searches were finally searched and aligned on the NCBI BLASTp webpage. Sequences were classified as present if the candidate ORFs returned good hits (e-value <1^−05^) to proteins of the same or closely related organism annotated with the exact function searched for. Sequences were classified as *potentially present* if: (1) only closely related proteins were identified using BLAST, but specific PFAM domains were present in the genome; (2) BLASTp returned good hits (e-value <1^−05^) to closely related functional proteins in any other organism. All other genes were classified as missing. The glycoprofiles were hierarchically clustered using seaborn cluster map with Euclidean method in Python. 

The computational analysis shown in Figure 3 supports the observation that glycosylation pathways differ significantly among diverse algal taxa, as all species investigated were predicted to have a unique combination of glycosylation enzymes. However, computational analysis only shows anticipated presence or absence of homologous enzymes and should always be supported by experimental analysis. For example, experimental analysis of glycosylation patterns of *P. purpureum* did not show activity of GnT I enzyme (Figure 1). However, GnT I is classified as present in the computational analysis. Similarly, all *O*-arabinosyltransferases from *A. thaliana* (HPAT1 and HPAT3) are classified as absent in *C. reinhardtii* (Figure 3), but core arabinose *O*-glycans have been detected in this microalga [124] (Figure 2). Nevertheless, computational analysis can give fundamental insight on microalgal species to be subsequently selected for further experimental analysis.

*C. reinhardtii* shows a very different enzyme population relative to humans and to *A. thaliana*, except for possible presence of an *O*-fucosyltransferase (POFUT1) that could result in core fucose *O*-glycans for this species. Another interesting result is the possible presence of a plant-like *N*-linked β(1,2)-xylosyltransferase (XYLT). *C. reinhardtii* is known to have xylose residues linked to the core GlcNAc and the antennas, and a previous study has demonstrated the presence of two xylosyltransferases, named XTA and XTB [129]. The absence of HPAT1 and HPAT3 in this microalga was unexpected, given that this microalga was shown to produce *O*-arabinose structures [124]. This suggests the potential presence of alternative enzymes in *C. reinhardtii* capable of linking core *O*-arabinoses.

Unlike *C. reinhardtii,* the green algae *B. braunii* and *C. vulgaris* show the potential presence of GnT I, which is supported by experimental analysis in *B. braunii*. Both species contain candidate genes for core α(1,3)-fucosyltransferase (FUT11), which is immunogenic in humans, and its presence strongly influences the efficacy of biopharmaceuticals [93]. *B. braunii* is the only species to show possible presence of α(2,6)-sialyltransferase (ST6GAL1), which is an important modification found in human proteins. The *O*-glycosylation pathway in *B. braunii* and *C. vulgaris* presents some similarities: the enzymes *O*-fucosyltransferase (POFUT1), and *O*-arabinosyltransferases from *A. thaliana* HPAT1 and HPAT3 are classified as possibly present in both species. However, the two species also show some differences. In fact, only *B. braunii* possibly presents homologues to the *O*-GalNAc transferase (GALNT1), whilst homologous enzymes of plant arabinosyltransferase (RRA1) are possibly present only in *C. vulgaris*.

*P. purpureum* shows the presence of GnT I, in contradiction with experimental data [108]. It also contains putative homologues to human α(1,6)-fucosyltransferase (FUT8) and FUT11, which can affect biopharmaceutical efficacy [4] and immunogenicity [93], respectively. *P. purpureum* is predicted to have a homologue to human β-1,4-galactosyltransferase 1 (B4GALT1), which is involved in the attachment of the fundamental human residue sialic acid. However, other enzymes involved in this pathway including α(2,3)-sialyltransferase, α(2,6)-sialyltransferase, GnT I, GnT IV, and GnT V were not detected. For *O*-glycans, *P. purpureum* shows the potential presence of human-like *O*-mannose and *O*-GalNAc cores, and homologous enzymes of plant arabinosyltransferase (RRA1). This enzyme, however, is not involved in core *O*-arabinose linkage; core *O*-arabinose enzymes are reported as missing.

Although both species belong to the stramenopile lineage, *P. tricornutum* and *N. gaditana* vary with respect to the presence of GnT I, which has been predicted in *P. tricornutum* but is not detected in *N. gaditana.* However, both species have putative homologues for human α(1,6)-fucosyltransferase and plant α(1,3)-fucosyltransferase, two enzymes linked to decreased efficacy of biopharmaceuticals. Additionally, *P. tricornutum* is also predicted to harbour a homologue to plant β(1,2)-xylosyltransferase (XYLT) that may add immunogenic residues, although the presence of this enzyme was not detected experimentally [109]. Regarding *O*-glycosylation, both species show possible presence of enzymes involved in the attachment of *O*-xylose and *O*-fucose cores. In addition, *P. tricornutum* may also produce *O*-GalNAc cores; however, experimental analysis of *O*-glycans produced in these species is still needed.

Although the computational analysis suggests that several model algal species possess promising characteristics for biopharmaceutical production, including the presence of GnT I, several potentially problematic enzymes are also present. Albeit antigenicity of recombinant biopharmaceuticals produced in microalgae has been tested (mostly to assess proper folding and showing mixed results) [57,59,64], immunogenicity of microalgal-based therapeutics has never been tested. However, given the diversity of microalgal glycans and the predicted differences with human glycosylation profiles, “humanization” of microalgae glycans via glyco-engineering is likely needed to elicit proper folding and remove possible immunogenic glycans, to lastly produce active and safe recombinant biopharmaceuticals from microalgae [4].

## 5. Strategies for Manipulating Protein Glycosylation 

In biofactories such as *E. coli*, yeasts, CHO cells, and plants, different glyco-engineering techniques have been successfully used to manipulate and “humanize” glycans to produce active and safe biopharmaceuticals for patients (*E. coli*: [130,131]. Yeasts: [132]. CHO cells: [133,134]. Plants: [79,135]). However, none of these techniques have yet been applied to microalgae. Glyco-engineering strategies can be divided in two main categories: protein engineering and cell engineering [97] (Figure 4). Protein glyco-engineering strategies target the recombinant glycoprotein before its translation (by modifying its DNA sequence), during its translation (by modifying its subcellular location), or after its translation (by modifying its glycosylation pattern) [97,136]. Cell glyco-engineering approaches introduce or modify the expression and the activity of target enzymes involved in the glycosylation pathways [97].

### 5.1. Protein Engineering

#### 5.1.1. Glycoprotein Sequence Engineering

This strategy is based on changing the amino acid sequence of a recombinant protein without changing its structure or function in the cell, to either (i) increase the number of glycans present or (ii) remove glycan attachment sites. The first strategy is used in the case of non-immunogenic recombinant glycoproteins in order to enhance their activity, while the second strategy prevents a protein from being glycosylated and therefore reduces its immunogenicity [137]. This approach was successful in CHO cells [146,147], and could be applied to microalgae, considering the advanced status of microalgal genetic manipulation in many different species [12]. However, this strategy presents many limitations, as altering a protein natural glycosylation is likely to cause diminished activity and stability, as demonstrated for IFN-ß [148] and for IgG-like antibody-based therapeutics [149]. 

#### 5.1.2. Subcellular Location Engineering

Although major protein modification occurs during ER and Golgi processing, glycans can be additionally modified during transport or within other organelles [112]. Targeting and transporting a recombinant protein to a specific organelle can result in a specific glycan pattern. For example, in plants, glycoproteins targeted and transported to vacuoles present characteristic paucimannosidic *N*-glycans [112]. Recombinant glucocerebrosidase (GCD), a biopharmaceutical used for Gaucher’s disease treatment, was found not to be effective unless it presented paucimannosidic *N*-glycans [79]. In vivo protein glyco-engineering has been achieved by expressing a recombinant GCD with a C-terminal vacuole-targeting signal in transgenic carrot cells, leading to the successful production of a paucimannosidic biopharmaceutical (ELELYSO^®^) [79], approved for Gaucher’s disease treatment by FDA in 2012 [2]. Microalgae, like plants, possess a vacuole, therefore it might be possible to target proteins to this organelle and obtain paucimannosidic *N*-glycans. Moreover, the microalga *P. tricornutum* naturally expresses paucimannosidic fucosylated *N*-glycans [109]. Although the function of microalgal vacuole is still under-characterized compared to higher plants [150], subcellular localization engineering might result in successful production of paucimannosidic biopharmaceuticals in microalgae. Interestingly, microalgal vacuole signal peptides for *P. tricornutum* have been identified [151]. However, subcellular location engineering presents a major downside: it is not possible to add/remove a targeted residue from the glycan. Therefore, this approach is strictly related to the organelle targeted for protein transportation and the resulting specific glycosylation pattern. Vacuole-targeting glyco-engineering, for example, is specific to proteins like GCD which require a paucimannosidic glycosylation pattern to be active. To remove a selected immunogenic residue, or to add a necessary residue, it is necessary to apply a different glyco-engineering approach.

#### 5.1.3. Glycosylation Pattern Engineering

Glycosylation pattern engineering targets the glycan after its synthesis by trimming it to its first residue and then reconstructing a new pattern in vitro. This post-translational remodelling of the glycan can be achieved either enzymatically or chemically. These two strategies are briefly described below, but for detailed insight we refer the reader to reviews by Wang and Lomino [138] and Chalker et al. [139].

Chemoenzymatic glycan remodelling uses enzymes to both trim the pre-existing glycan and to construct the in vitro desired pattern [97]. Studies focusing on the enzymatic single step transfer of a pre-assembled glycan have shown successful results in *E. coli* [130,131], *P. pastoris* [152] and CHO cells [153].

During chemoselective and site-specific glycosylation, a tag is inserted on glycosylation sites on the amino acid backbone by site-directed mutagenesis. A glycan is then added to these tags by bio-orthogonal chemoselective ligation with a modified glycan carrying a compatible functional group. The tag on the protein and the functional group on the glycan are complementary and will selectively recognize each other, enabling a targeted insertion of glycans onto the amino acid backbone [138,139] (Figure 4A). Glycosylation pattern engineering is independent of the expression host and therefore could be used in microalgae.

### 5.2. Cell Glyco-Engineering

To produce desired human-like glycans or add crucial human residues to existing glycans, it is possible to target the species-specific glycosylation enzymes responsible for the attachment of immunogenic glycans [20,21,22,23,24] using cell glyco-engineering approaches. Cell glyco-engineering can be achieved by using inhibitors or genetic engineering to introduce or modify the expression of enzymes involved in glycosylation pathways [97,138] (Figure 4B).

#### 5.2.1. Glyco-Engineering by Inhibitor Interference

An inhibitor interference approach uses small molecules that specifically inhibit or interfere with the activity of specific enzymes in the glycosylation pathway [138] (Figure 4B). Small molecules such as N-butyl deoxynojirimycin, kifunensine, and swainsonine inhibit the activity of glycosylation enzymes such as the ER α-glucosidases I and II, the ER α-mannosidase-I and the Golgi α-mannosidase II. This technique could be applied to microalgae if non-toxic inhibitors specific to microalgal glycosylation enzymes (such as fucosyltransferase and xylosyltransferase) can be identified. However, successful inhibitors might be difficult to find for these organisms, considering the evolutionary and metabolic biodiversity of microalgae. One example is the insensitivity of the diatom *P. tricornutum* to the activity of terbinafine (well-known inhibitor of the ubiquitous enzyme squalene epoxidase) [154,155]. Moreover, inhibitor interference strategies can only prevent the attachment of specific residues, they cannot enhance or introduce expression of human glycosylation enzymes absent in microalgae (such as the enzyme GnT I in *C. reinhardtii* and *P. purpureum*). To obtain “hybrid” or “complex” glycans it is necessary to genetically engineer the glycosylation machinery of the host organism.

#### 5.2.2. Genetic Glyco-Engineering

Genetic glyco-engineering can be achieved by either introduction of heterologous glycosylation machinery or inactivation of endogenous enzymes. The integration of one or more genes coding glyco-enzymes can be achieved by random insertion or targeted knock-in (KI) of recombinant DNA into the host genome. The inserted DNA encodes specific enzymes absent in the wild type organism that will add the desired residues to the recombinant glycoprotein to obtain properly folded, active, and safe recombinant biopharmaceuticals. Although expression of exogenous enzymes has been proven as an effective strategy in CHO cells [140,156,157], expression must be carefully regulated. Overexpression of glycosylation enzymes can lead to the attachment of extra residues on the glycan, possibly affecting the stability and activity of the recombinant protein.

A major technology advancement across all host systems including microalgae has been the development of targeted genome engineering aided by DNA nucleases, including zinc finger nucleases (ZFNs) [141], transcription activator–like effector nucleases (TALENs) [142], and clustered regularly interspaced short palindromic repeat/targeted Cas9 endonuclease (CRISPR-Cas9) [143,144,145] (Figure 4B). DNA nucleases are able to generate double stranded DNA breaks at precise genomic locations of interest, increasing the likelihood of exogenous DNA integration at a desired location, instead of at a random location. Random integration, while much easier to achieve following conventional DNA transformation strategies, is prone to position effect regarding transgene expression and uncharacterised genomic disruptions. Targeted gene integration can improve transgene expression levels when appropriate integration locations are selected; namely, regions that permit insertion of exogenous genes without disrupting the host natural gene expression whilst simultaneously lowering the risk of silencing of the exogenous DNA. Such regions, known as “safe harbours” have been frequently used in human and mouse cell lines [158,159,160]. Identified safe harbour loci knowledge, coupled with efficient endonuclease-mediated targeted integration transformation protocols, can revolutionise genetic engineering strategies to produce recombinant biopharmaceuticals by circumventing reproducibility and stability issues associated with random chromosomal integration [161].

Similar to exogenous enzyme insertion, endogenous enzyme removal strategies can be random or targeted. Gene(s) knock-down (KD) strategies result in reduced activity of the glycosylation enzymes and can be achieved by gene(s) silencing techniques, such as RNA interference [162] or CRISPR interference (CRISPRi) [163]. However, gene(s) KD does not completely inactivate enzyme expression, often resulting in still high levels of the targeted undesired enzyme activity and consequently in the attachment of the unwanted immunogenic residues. Gene(s) knock-out (KO), on the other hand, disrupts a gene that encodes a specific enzyme in the glycosylation pathway in order to permanently suppress its function [138]. Gene KO can be obtained by targeted genome editing strategies and/or random insertional mutagenesis [164,165]. This strategy results in loss-of-function mutant cell lines that do not produce the target enzyme, and consequently, lack the undesired residues [97]. Targeted gene(s) KO strategies (ZFNs and CRISPR-Cas9) were successfully used in CHO cells to inactivate the gene responsible for core-linked fucose residues (*FUT8*) and generate non-fucosylated glycans [133,134]. Non-fucosylated biopharmaceuticals expressed in *FUT8*-KO cell lines showed increased efficacy [4]. Mogamulizumab (POTELIGEO^®^) and Benralizumab (MEDI-563, Fasenra^TM^) are two examples of FDA approved non-fucosylated biopharmaceuticals expressed in *FUT8*-KO CHO cell lines [166]. However, inactivating gene(s) might have secondary detrimental effects on native proteins and consequently on the host organism.

In non-mammalian hosts, integrating human glycosylation enzymes is not sufficient to avoid the presence of immunogenic residues. Exogenous gene integration needs to be coupled with KD or KO of endogenous gene(s) coding for species-specific enzymes [97,138]. A combined approach of exogenous gene insertion and inactivation of endogenous genes was successfully used in yeasts [132,167] and plants species such as *Lemna minor* [168], *A. thaliana* [135,169] and *N. benthamiana* [170,171,172].

## 6. Future Perspective for Glyco-Engineering in Microalgae 

The effectiveness of glyco-engineering approaches has been demonstrated in many different organisms, both mammalian and non-mammalian, setting the stage for glyco-engineering in microalgae. However, to progress glyco-engineering strategies in microalgae, it is still necessary to improve our knowledge of microalgal glycosylation status (by genomic and experimental evidence) and thus generate a comprehensive and detailed overview of the glycosylation pathways. Unlike in common expression systems, the *N*- and *O*-glycosylation pathways in microalgae have yet to be fully characterized. In this review, we provide an in silico analysis of glycosylation enzymes in five microalgal species, based on homology with higher organisms (Figure 3). These reconstructed pathways, combined with previous glycosylation analysis in other microalgal species (both empirical and putative), support the presence of a vast diversity of glycosylation patterns and can be used to identify future targets for glyco-engineering in microalgae.

An important consideration for recombinant biopharmaceutical production in microalgae is the choice of host species, as glycosylation status among microalgae has been shown to be very diverse (Section 4.1 and Section 4.2). For *N*-glycans, a first selection criteria might be the presence or absence of the GnT I enzyme. GnT I-independent species show complete absence of “hybrid” and “complex” human-like glycans, whereas they still present undesired enzymes and their products (namely Xyl, Fuc, and methylation of residues). However, performing a KI of the GnT I enzyme in a GnT I-independent species lacking FuT, XyT, and MeT (based on computational analysis) might result in non-immunogenic “hybrid” and “complex” glycans. A recent study pursued this strategy in *C. vulgaris* and characterised “hybrid” *N*-glycans lacking fucose or xylose residues after overexpression of recombinant GnT I [111]. Alternatively, GnT I-dependent species already present “hybrid” and “complex” glycans. However, FuT, XyT, and MeT enzymes may still be present and active. Selection of a species that does not possess FuT, XyT, and MeT, or targeted KD or KO of these enzymes would prevent the attachment of the immunogenic residues. Species such as *B. braunii* and *P. tricornutum* are GnT I-dependent species lacking FuT, XyT, and MeT that also have a putative α-mannosidase II (computational analysis). These are particularly interesting candidates to produce paucimannosidic non-immunogenic glycans. However, this strategy is limited by native methylation of residues in *B. braunii*, a possibly immunogenic trait [110]. *P. tricornutum*, on the other hand, does not present methylated residues, but has a core α(1,3)-fucosyltransferase. It might be effective to KD or KO the fucosyltransferase in *P. tricornutum* and obtain paucimannosidic non-immunogenic glycans in an already well-known and well-characterised microalgal species. 

Both GnT I-independent and GnT I-dependent species are missing a very important residue present in humans: sialic acid. Enzymes such as α(2,3)-sialyltransferase (ST3GAL3), α(2,6)-sialyltransferase (ST6GAL1), β(1,4)-galactosyltransferase (B4GALT1), GnT I, GnT IV, and GnT V are all involved in the sialic acid pathway, and completely (or almost completely) absent in microalgae. It has been shown how overexpression of this pathway in CHO cells resulted in improved recombinant biopharmaceuticals production [173,174,175]. Figure 3 shows putative presence of some of these enzymes in some microalgal species. However, the complete pathway is absent in all the species that were computationally analysed. Furthermore, sialic acid has never been reported in any of the species experimentally analysed. Therefore, an extensive amount of glyco-engineering might be required to obtain this residue in any of these species.

The absence of a consensus sequence, a common core, and recurring patterns has made a comprehensive *O*-glycosylation immunogenicity analysis more challenging. The only glycosylation analysis available in microalgae (for *C. reinhardtii*) showed similarity with higher plants, with the addition of methylated residues. Immunogenicity of recombinant *O*-glycoproteins from plants has not been tested, however the absence of a similar *O*-glycan core in humans inspires little confidence. The computational analysis shown in Figure 3 offers interesting insight. Human *O*-xylose and *O*-fucose cores might be present in more than one microalgal species. Interestingly, microalgae and plants show limited homology (Figure 3). However, this should be considered only as a starting point for further experimental analysis of microalgal *O*-glycosylation.

Given the vast diversity of glycosylation status among microalgal species and considering the existing variety of glycans and their impact on different classes of biopharmaceuticals, it is difficult to predict a precise number of gene(s) or glyco-engineering approaches to obtain a fully “humanised” recombinant biopharmaceutical. However, based on the experimental and computational information collected, and the different successful glyco-engineering approaches utilised in other species, we postulate that producing a vacuole-targeted glucocerebrosidase in a *P. tricornutum* fucosyltransferase-KO cell line could be the fastest way to obtain the first commercialised recombinant biopharmaceutical from microalgae. 

In conclusion, there are still major hurdles limiting glyco-engineering in microalgae. Obtaining experimental information for a single species is challenging and may very well expose immunogenic traits in the selected microalgal glycans. Some immunogenic traits may be compatible with protein or cell engineering, while others may be too complex or widespread to warrant further investigation. There is also no clear evidence to guide which microalgal species might be more appropriate in this regard over others. Moreover, considering the diversity of glycosylation among microalgae, the information obtained cannot be extended to other species, and each microalga must be experimentally investigated. However, glycosylation status of microalgae is still closer to human patterns when compared to glycosylation in bacteria and yeasts. Given the recent advancements in various model microalgal genetic toolkits, glyco-engineering approaches hold great potential to produce non-immunogenic biopharmaceuticals in microalgae.

## Figures and Tables

**Figure 1 cells-09-00633-f001:**
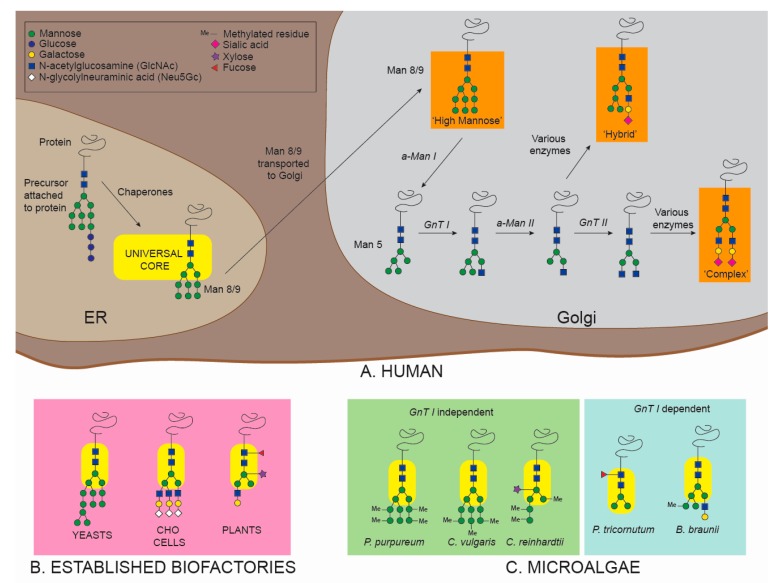
A schematic of *N*-glycosylation patterns found in (**A**) humans, (**B**) established biofactories including yeasts, plants, CHO cells, and (**C**) specific *N*-glycosylation patterns in microalgae. Glycans shown for (**A**) humans and (**B**) established biofactories are only illustrative and do not represent the totality of possible glycoforms obtained from these organisms. Differences in specificity of yeast, CHO, plant, and microalgal Golgi glycosyltransferases and glycosidases lead to variations in the final glycosylation profiles compared to humans. Consequently, glycans *N*-linked to recombinant proteins produced in these biofactories differ from native human proteins, necessitating glycan engineering to produce efficient and safe biopharmaceuticals in these alternative host systems. Green circle = Mannose. Blue circle = Glucose. Yellow circle = Galactose. Blue square = GlcNAc. White diamond = Neu5Gc. Me = Methylated residue. Fuchsia diamond = Sialic acid. Purple star = Xylose. Red triangle = Fucose. α-Man I = α-mannosidase I. GnT I = *N*-acetylglucosaminyltransferase I. α-Man II = α-mannosidase II. GnT II = *N*-acetylglucosaminyltransferase II.

**Figure 2 cells-09-00633-f002:**
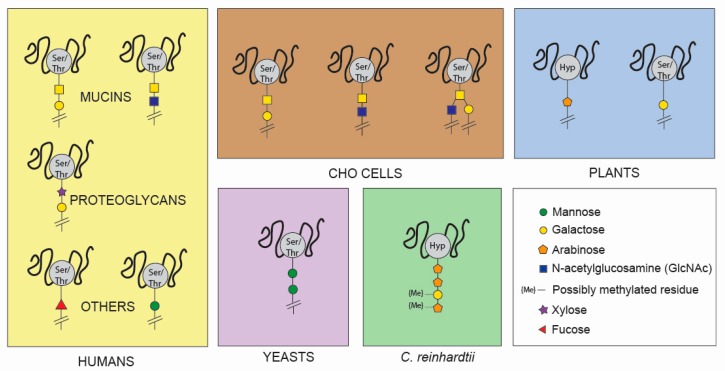
A comparison of the different *O*-glycosylation patterns among humans, CHO cells, yeasts, plants, and microalgae. Experimental evidence for *O*-glycosylation in microalgae is limited to *C. reinhardtii* [124]. Native *C. reinhardtii* proteins possess a (Hyp-*O*-Ara-Ara) core and methylated residues, characteristics that differ significantly from human *O*-glycosylation patterns. Ser/Thr = serine or threonine. Hyp = hydroxyproline. Green circle = Mannose. Yellow circle = Galactose. Orange pentagon = Arabinose. Blue square = GlcNAc. Me = Methylated residue. Purple star = Xylose. Red triangle = Fucose.

**Figure 3 cells-09-00633-f003:**
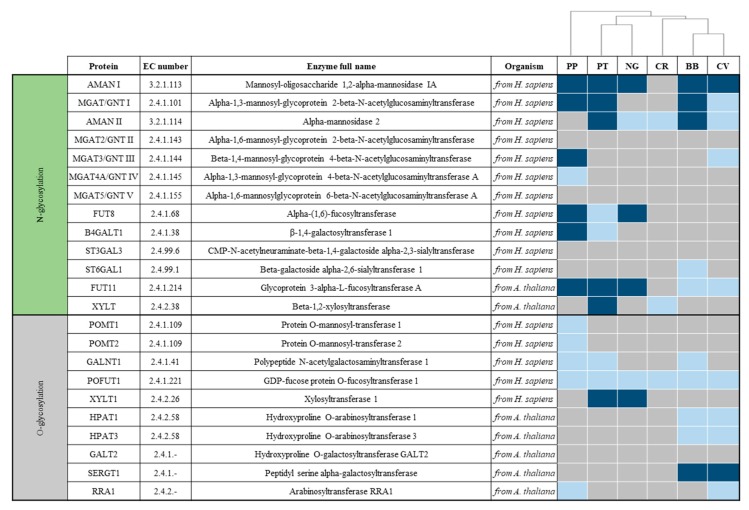
Hypothetical presence of protein *N*-glycosylation and *O*-glycosylation enzymes in the representative microalgal species, *P. purpureum* (PP), *P. tricornutum* (PT), *Nannochloropsis gaditana* (NG), *C. reinhardtii* (CR), *B. braunii* (BB), and *C. vulgaris* (CV), compared with glycosylation enzymes from *H. sapiens* and *A. thaliana*. Enzymes are classified as *present* (dark blue)*, potentially present* (light blue) or *missing* (grey).

**Figure 4 cells-09-00633-f004:**
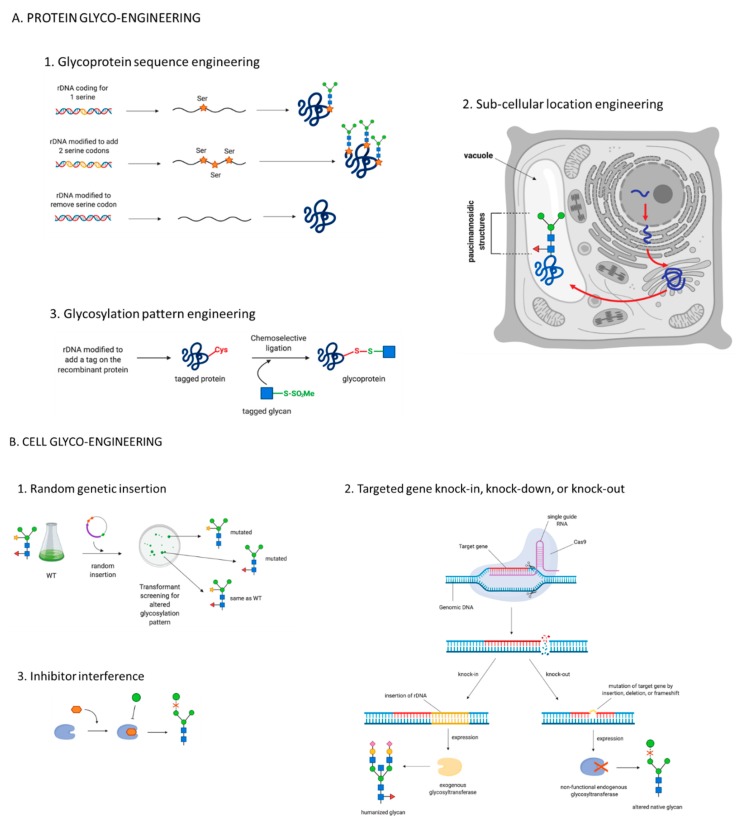
A schematic of different glyco-engineering strategies; (**A**) protein engineering and (**B**) cell engineering. Protein glyco-engineering approaches (**A**) can target (1) the recombinant DNA sequence (rDNA) [137], (2) the sub-cellular location of the biopharmaceutical [112], and (3) the glycosylation pattern of the translated protein [138,139]. Cell glyco-engineering strategies (**B**) can modify the activity of glycosylation enzymes by (1) random genetic insertion [140], (2) targeted gene knock-in or knock-out [141,142,143,144,145], and (3) inhibitor interference [138]. Green circle = Mannose. Blue circle = Glucose. Yellow circle = Galactose. Blue square = GlcNAc. Fuchsia diamond = Sialic acid. Yellow star = Xylose. Red triangle = Fucose.

**Table 1 cells-09-00633-t001:** Twenty years of recombinant biopharmaceutical production in microalgae.

Organism	Organelle	Protein	Reference
*C. reinhardtii*	Chloroplast	E7 of HPV-16	[48]
		D2-CTB	[49]
		α-galactosidase	[50]
		Phytase	[50]
		Xylanase	[50]
		Pfs25	[51]
		Pfs28	[51]
		Pfs25-CTB	[52]
		E2	[53]
		Pfs48/45	[54]
		M-SAA	[55]
		Anti-HSV glycoprotein D Isc	[56]
		12FN3	[57]
		Erythropoietin	[57]
		HMGB1	[57]
		Interferon β	[57]
		Proinsulin	[57]
		SAA-10FN3	[57]
		VEGF	[57]
		Allophycocyanin	[58]
		VP1-CTB	[59]
		V28	[60]
		Anti-PA 83 anthrax IgG1	[61]
		Anti-CD22-gelonin sc	[62]
		Anti-CD22-ETA sc	[63]
		GAD65	[64]
		TRAIL	[65]
		Phytase (AppA)	[66]
		Metallothionein-2	[67]
*C. reinhardtii*	Nucleus	Human Epidermal Growth Factor	[68]
		VEGF-165	[69]
		GBSS-AMA1	[70]
		GBSS-MSP1	[70]
		Erythropoietin	[42]
		Sep-15	[71]
		Lolium Perenme IBP	[72]
		β-1,4-endoxylanase	[73]
*C. vulgaris*	Nucleus	Human growth hormone	[14]
*C. sorokiniana*			
*C. ellipsoidea*	Nucleus	mNP-1	[15]
		NP-1	[74]
		Flounder growth hormone	[75]
*D. salina*	Chloroplast	α-galactosidase	[50]
		Phytase	[50]
		Xylanase	[50]
*D. salina*	Nucleus	V28	[76]
		HBsAg	[16]
*P. tricornutum*	Nucleus	Anti-Hepatitis B IgG	[19]
		Anti-MARV NP IgG	[77]
*N. oculata*	Nucleus	Bovine lactoferricin (LFB)	[18]
	Nucleus	Flounder growth hormone	[17]

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
