# Peer review of "Perspectives for Glyco-Engineering of Recombinant Biopharmaceuticals from Microalgae"

_cells, 2020, doi:10.3390/cells9030633_

Round 1
Reviewer 1 Report
This is a good review, interesting to read by both glycobiologists and biotechnologists and non-specialists. I believe that this requires minor refinement. My recommendations: 1) For one recombinant glycoprotein obtained in algal cells, present the structures of a complete set of isolated glycans. Preferably, a glycoprotein having both N- and O-chains, preferably closest to the human counterpart with respect to glycosylation. This information may be provided as Supplementary material. 2) The article discusses the potential immunogenicity of incorrectly-glycosylated proteins. It would be nice to give experimental data on antigenicity. Of course, immunogenicity cannot be studied at the current stage of development, but antigenicity, interaction with natural (pre-existing) antibodies in vitro can be studied, and there is no doubt that this has already been investigated. 3) How many genes (total) must be transfected / knocked out from algae in order to create a fully humanized glycoprotein, according to the author? 4) It would be nice to review data on folding of algae-derived glycoproteins. 5) Minor remarks 1. Fig. 1 and 4 there is no guide for converting monosaccharide symbols (done in the form of a ball/rectangle) to common notation. Fig. 2 contains it, but not for all monosaccharides. 2. At page 38 "glycol" should be corrected.Author Response
REVIEWER 1
This is a good review, interesting to read by both glycobiologists and biotechnologists and non-specialists. I believe that this requires minor refinement.
We would like to thank the reviewer for her/his very positive feedback on the manuscript.
My recommendations: 1) For one recombinant glycoprotein obtained in algal cells, present the structures of a complete set of isolated glycans. Preferably, a glycoprotein having both N- and O-chains, preferably closest to the human counterpart with respect to glycosylation. This information may be provided as Supplementary material.
While we agree with reviewer 1 that this information would be very interesting, unfortunately, there is no glycosylation analysis published or available so far for glycan patterns of recombinant proteins in microalgae, therefore such a comparison is not possible.
2) The article discusses the potential immunogenicity of incorrectly-glycosylated proteins. It would be nice to give experimental data on antigenicity. Of course, immunogenicity cannot be studied at the current stage of development, but antigenicity, interaction with natural (pre-existing) antibodies in vitro can be studied, and there is no doubt that this has already been investigated.
The difference between antigenicity and immunogenicity and their analysis for recombinant protein production is very important, and we would like to thank the reviewer for suggesting the addition of this subject. We broaden the description of glycosylation implication in folding and immunogenicity, also adding antigenicity.
Line 245, page 12: “Glycosylation significantly enhances yield, folding, efficacy, and pharmacokinetics of recombinant biopharmaceuticals [30]. During recombinant protein production, non-human host organisms can attach glycan residues (monosaccharides) that would be absent on the human endogenous protein, potentially resulting in lower yields, incorrect folding, and inefficacy of the biopharmaceutical [10]. For example, glycosylation plays a fundamental role in activity of antibody-based therapeutics [88]. In fact, antibody are able to target and kill hostile cells by antibody-dependent cellular cytotoxicity (ADCC) or complement-dependent cytotoxicity (CDC) mechanisms, and both ADCC and CDC are directly related to glycosylation presence and status of the antibody [89-91]. Therefore, an antibody carrying incorrect glycosylation might show diminished activity. Moreover, correct glycosylation and proper folding also affect immunogenicity and antigenicity of recombinant biopharmaceuticals. Immunogenicity is the capability of a molecule to trigger an immune response in the patient, whilst antigenicity is the ability of a molecule to bind immune system products. Therefore, an antigen is not necessarily an immunogen, whilst an immunogen is inevitably an antigen. Immunogenic biopharmaceuticals can trigger an immune response in the patient resulting in accelerating clearance during therapy or, in some rare cases, life threatening complications [92]. Glycans can trigger an immunogenic reaction either indirectly or directly. Glycosylation can have an indirect effect on immunogenicity by influencing therapeutic proteins folding, solubility and structural stability [93]. In fact, incorrect or no glycosylation can alter the secondary/tertiary structure and/or prompt aggregation of therapeutic proteins, factors breaking the immune tolerance of the patient. Moreover, specific non-human residues can be directly recognised as exogenous by the patient immune system and trigger an immunogenic response [94]. At least four non-human glycans have been identified as being able to induce an immune response in humans. These residues are: α-Gal, Neu5Gc, β(1,2)-xylose and α(1,3)-fucose [93]. The α-Gal and Neu5Gc residues are present in therapeutics produced in mammalian cells such as CHO cells, while β(1,2)-xylose and α(1,3)-fucose are present in plant and microalgal-produced glycoproteins [93]. Given the crucial role of glycosylation in folding, activity, and immunogenicity of biopharmaceuticals, it is essential to understand the glycosylation capabilities of a chosen biofactory to produce properly folded, effective, and safe recombinant therapeutics”.
Line 530, page 24: “Although the computational analysis suggests that several model algal species possess promising characteristics for biopharmaceutical production, including the presence of GnT I, several potentially problematic enzymes are also present. Albeit antigenicity of recombinant biopharmaceuticals produced in microalgae has been tested (mostly to assess proper folding and showing mixed results) [57,59,64], immunogenicity of microalgal-based therapeutics has never been tested. However, given the diversity of microalgal glycans and the predicted differences with human glycosylation profiles, “humanization” of microalgae glycans via glyco-engineering is likely needed to elicit proper folding and remove possible immunogenic glycans, to lastly produce active and safe recombinant biopharmaceuticals from microalgae [4]”.
3) How many genes (total) must be transfected / knocked out from algae in order to create a fully humanized glycoprotein, according to the author?
Considering the complexity of the subject, and all the variables involved, it is difficult to hypothesise a specific number to obtain a humanised glycoprotein. However, we tried to describe what we think would be the fastest way to obtain the first commercialised recombinant biopharmaceutical from microalgae.
Line 783, page 34: “Given the vast diversity of glycosylation status among microalgal species, and considering the existing variety of glycans and their impact on different classes of biopharmaceuticals, it is difficult to predict a precise number of gene(s) or glyco-engineering approaches to obtain a fully “humanised” recombinant biopharmaceutical. However, based on the experimental and computational information collected, and the different successful glyco-engineering approaches utilised in other species, we postulate that producing a vacuole-targeted glucocerebrosidase in a P. tricornutum fucosyltransferase-KO cell line could be the fastest way to obtain the first commercialised recombinant biopharmaceutical from microalgae”.
4) It would be nice to review data on folding of algae-derived glycoproteins.
The analysis of antigenicity is mainly based on correct folding of the protein (also correlated to correct/incorrect glycosylation), therefore in the part we added for comment #2, there is also a small part citing three papers that analysed folding of algae-derived glycoproteins (via antigenicity).
Line 530, page 24: “Although the computational analysis suggests that several model algal species possess promising characteristics for biopharmaceutical production, including the presence of GnT I, several potentially problematic enzymes are also present. Albeit antigenicity of recombinant biopharmaceuticals produced in microalgae has been tested (mostly to assess proper folding and showing mixed results) [57,59,64], immunogenicity of microalgal-based therapeutics has never been tested. However, given the diversity of microalgal glycans and the predicted differences with human glycosylation profiles, “humanization” of microalgae glycans via glyco-engineering is likely needed to elicit proper folding and remove possible immunogenic glycans, to lastly produce active and safe recombinant biopharmaceuticals from microalgae [4]”.
5) Minor remarks
1. Fig. 1 and 4 there is no guide for converting monosaccharide symbols (done in the form of a ball/rectangle) to common notation. Fig. 2 contains it, but not for all monosaccharides.
We added the guide in the figure captions (page 15 and 20) as recommended by reviewer 1.
- At page 38 "glycol" should be corrected.
We corrected the word “Glycol” with “Protein” as recommended by reviewer 1.

Reviewer 2 Report
Barolo L et al.: PERSPECTIVES FOR GLYCO-ENGINEERING OF RECOMBINANT BIOPHARMACEUTICALS FROM MICROALGAE
An interesting and relevant topic.
In my opinion, this review lacks a bit of focus. On the one hand, it tries to focus on glycoengineering in a microalgae system in the context of known principles, on the other hand it branches out into details (but not well described) of e.g. CRISPR. I would rather shorten the explanations on technologies that can be used in microalgae (this is not a review on those methods) and cite relevant literature in case the reader wants to know details about them.
In chapters 2. and 3. I would suggest to include some examples of biopharmaceuticals from alternative hosts that made it to the market (unfortunately very few) and discuss reasons for that. There already is a wealth of publications about the great potential of plant, algae … systems for biopharmaceutical production (see table 1: “Twenty years of recombinant biopharmaceutical production (!) in microalgae”) – but where are they on a real / large scale?
Some details:
Introduction:
Please correct: Serum is not used in CHO processes (any more). In fact, it is extremely unlikely that a process will get regulatory clearance if it incorporates any animal derived components nowadays. The information in the manuscript was derived from papers 9 and 10, both not primarily on CHO cells.
Tractability of C.reinhardtii: I do not understand?
P7, Ebola: Was the ZMapp really produced in N.benthamiana to treat Ebola in the Congo (in humans)? [35] is about scientific studies, not production?
P9, first paragraph: Without wanting to be too picky: Expression is not in the nucleus but transcription. Maybe better “expression from the nucleus”?
What is meant by “adding specific pre-peptides to the recombinant aa sequence”? I guess you mean signal peptides?
P13: The cellular importance of glycosylation is not the main focus here (lines 2-4) but its importance for the biopharmaceutical protein other than immunogenicity (ADCC, …). Maybe elaborate a bit more on that (lines 5-6); see e.g. Jeffries R 2009, Nature Reviews Drug Discovery 8, 226-234
Fig 1: The figure suggests one specific glycosylation pattern for CHO – this is not the case! Please amend. By far not all N-glycosylation looks like this, especially in MABs. Also, please include the enzyme abbrevations in the legend.
P26: Remove (Baiet et al. 2011; also on P30 [103]
References: Why sometimes hyperlinks?
Fig3 and methods described therein: Reference is lacking. Or is this your own data? Then maybe this is something you want to publish, but not in a review?! Better ask the editor about that.
Fig4: This is a huge figure with lots of details which either need to be explained or removed. E.g. B.3. sits there without any explanation. E.g. B.2.: It is not purposeful to cram CRISPR into this figure. And so on…
P28/29: Contradiction: “This strategy…without changing its .. activity… The first strategy is used … to enhance their activity…” ? The activity of a protein can be affected by both, its aa composition and its glycosylation.
P29: What is the difference between reducing and removing a glycosylation site?
P32, 5.2.: Once more, here and throughout the manuscript: glycoengineering is not only about reducing immunogenicity, but e.g. also about activity (see e.g. rise in ADCC in the case of non-fucosylated Herceptin)
P33: [136] and papers cited therein does, as far as I think I understood, not describe the use of inhibitors in biopharmaceutical production in CHO but only in small scale experiments. Right? In that case, your sentence would be misleading.
P37: “Both… dependent [what?] are missing a …”
Author Response
An interesting and relevant topic.
We would like to thank the reviewer for her/his very positive feedback on the manuscript.
In my opinion, this review lacks a bit of focus. On the one hand, it tries to focus on glycoengineering in a microalgae system in the context of known principles, on the other hand it branches out into details (but not well described) of e.g. CRISPR. I would rather shorten the explanations on technologies that can be used in microalgae (this is not a review on those methods) and cite relevant literature in case the reader wants to know details about them.
We agree with reviewer 2 that the first version of the manuscript was lacking focus, therefore as recommended, we have shorten Section on technologies significantly (Section 5 STRATEGIES FOR MANIPULATING PROTEIN GLYCOSYLATION), to focus the review on the relevant topic of the application of glyco-engineering strategies in microalgae.
Line 574, page 26: “This strategy is based on changing the amino acid sequence of a recombinant protein without changing its structure or role in the cell, to either (i) increase the number of glycans present or (ii) remove glycan attachment sites. The first strategy is used in the case of non-immunogenic recombinant glycoproteins in order to enhance their activity, while the second strategy prevents a protein from being glycosylated and therefore reduces its immunogenicity [138]. This approach was successful in CHO cells [147,148], and could be applied to microalgae, considering the advanced status of microalgal genetic manipulation in many different species [12]. However, this strategy presents many limitations, as altering a protein natural glycosylation is likely to cause diminished activity and stability, as demonstrated for IFN-ß [149] and for IgG-like antibody-based therapeutics [150]”.
Line 603, page 27: “In vivo protein glyco-engineering has been achieved by expressing a recombinant GCD with a C-terminal vacuole-targeting signal in transgenic carrot cells, leading to the successful production of a paucimannosidic biopharmaceutical (ELELYSO®) [79], approved for Gaucher’s disease treatment by FDA in 2012 [2]. Microalgae, like plants, possess a vacuole, therefore it might be possible to target proteins to this organelle and obtain paucimannosidic N-glycans”.
Line 654, page 29: “Small molecules such as N-butyl deoxynojirimycin, kifunensine, and swainsonine inhibit the activity of glycosylation enzymes such as the ER α-glucosidases I and II, the ER α-mannosidase-I and the Golgi α-mannosidase II. This technique could be applied to microalgae assuming that non-toxic inhibitors specific to microalgal glycosylation enzymes (such as fucosyltransferase and xylosyltransferase) can be identified”.
In chapters 2. and 3. I would suggest to include some examples of biopharmaceuticals from alternative hosts that made it to the market (unfortunately very few) and discuss reasons for that. There already is a wealth of publications about the great potential of plant, algae … systems for biopharmaceutical production (see table 1: “Twenty years of recombinant biopharmaceutical production (!) in microalgae”) – but where are they on a real / large scale?
As already highlighted by the reviewer 2, unfortunately the number of commercialised biopharmaceuticals produced from alternative hosts is very low, and does not include microalgal species. Anyway, we added the only two examples of approved biopharmaceuticals from alternative hosts that we are aware of.
Line 205, page 11: “Unfortunately, only an exiguous amount of recombinant biopharmaceuticals produced in alternative host systems successfully passed clinical trials and was commercialised [2,78]. One is ELELYSO®, a recombinant glucocerebrosidase produced in carrot cells and approved for Gaucher’s disease treatment by FDA in 2012 [2,79]. The second one is ZMapp, an antibody cocktail administered during the Ebola virus outbreak in Western Africa [36-38]. One component of ZMapp (cZMAb) was produced in N. benthamiana [37,38]. Recently, two new antibody therapies outperformed ZMapp, leading to a different approved therapy for the Ebola virus [80]. Microalgal biopharmaceuticals are still absent from the market, mainly due to the low yields of recombinant proteins obtained from these biofactories [46,47]”.
Some details:
Introduction:
Please correct: Serum is not used in CHO processes (any more). In fact, it is extremely unlikely that a process will get regulatory clearance if it incorporates any animal derived components nowadays. The information in the manuscript was derived from papers 9 and 10, both not primarily on CHO cells.
We deleted that part as recommended by reviewer 2.
Line 90, page 5: “Biopharmaceutical production with CHO cells is expensive because of the complex culturing requirements associated, difficult to scale, and susceptible to contamination with human viruses and prions [9,10]”.
Tractability of C.reinhardtii: I do not understand?
We changed that part.
Line 98, page 5: “due to its advantageous biological features [12,13]”.
P7, Ebola: Was the ZMapp really produced in N.benthamiana to treat Ebola in the Congo (in humans)? [35] is about scientific studies, not production?
We missed a couple of citations, to better explain the production of ZMapp. We added two citations in lane 145, page 7.
- Budzianowski, J. Tobacco against Ebola virus disease. Przegl Lek 2015, 72, 567-571.
- Davidson, E.; Bryan, C.; Fong, R.H.; Barnes, T.; Pfaff, J.M.; Mabila, M.; Rucker, J.B.; Doranz, B.J. Mechanism of Binding to Ebola Virus Glycoprotein by the ZMapp, ZMAb, and MB-003 Cocktail Antibodies. J Virol 2015, 89, 10982-10992, doi:10.1128/JVI.01490-15.
Moreover, we also added another part where we explain which component of ZMapp was produced in N. benthamiana, and we also explain which therapy is nowadays utilised.
Line 208, page 10: “The second one is ZMapp, an antibody cocktail administered during the Ebola virus outbreak in Western Africa [36-38]. One component of ZMapp (cZMAb) was produced in N. benthamiana [37,38]. Recently, two new antibody therapies outperformed ZMapp, leading to a different approved therapy for the Ebola virus [80]”.
P9, first paragraph: Without wanting to be too picky: Expression is not in the nucleus but transcription. Maybe better “expression from the nucleus”?
We changed that, line 189 page 9.
What is meant by “adding specific pre-peptides to the recombinant aa sequence”? I guess you mean signal peptides?
We changed it into “signal”, line 190 page 9.
P13: The cellular importance of glycosylation is not the main focus here (lines 2-4) but its importance for the biopharmaceutical protein other than immunogenicity (ADCC, …). Maybe elaborate a bit more on that (lines 5-6); see e.g. Jeffries R 2009, Nature Reviews Drug Discovery 8, 226-234
This comments relates to reviewer 1 ‘comment, we broaden the description of glycosylation implication in folding and immunogenicity, also adding its fundamental role in antibody activity as suggested by both reviewers
Line 245, page 11: “Glycosylation significantly enhances yield, folding, efficacy, and pharmacokinetics of recombinant biopharmaceuticals [30]. During recombinant protein production, non-human host organisms can attach glycan residues (monosaccharides) that would be absent on the human endogenous protein, potentially resulting in lower yields, incorrect folding, and inefficacy of the biopharmaceutical [10]. For example, glycosylation plays a fundamental role in activity of antibody-based therapeutics [88]. In fact, antibody are able to target and kill hostile cells by antibody-dependent cellular cytotoxicity (ADCC) or complement-dependent cytotoxicity (CDC) mechanisms, and both ADCC and CDC are directly related to glycosylation presence and status of the antibody [89-91]. Therefore, an antibody carrying incorrect glycosylation might show diminished activity.
Fig 1: The figure suggests one specific glycosylation pattern for CHO – this is not the case! Please amend. By far not all N-glycosylation looks like this, especially in MABs. Also, please include the enzyme abbrevations in the legend.
The patterns reported in figure 1 were only illustrative of possible immunogenicity of glycans in different species. We modified the caption accordingly, to further highlight this.
Line 318, page 15: “Figure 1. A schematic of N-glycosylation patterns found in (A) humans, (B) established biofactories including yeasts, plants, CHO cells, and (C) specific N-glycosylation patterns in microalgae. Glycans shown for (A) humans and (B) established biofactories are only illustrative and do not represent the totality of possible glycoforms obtained from these organisms. Differences in specificity of yeast, CHO, plant, and microalgal Golgi glycosyltransferases and glycosidases lead to variations in the final glycosylation profiles compared to humans”.
We have also included the enzyme abbreviations in the legend as recommended.
P26: Remove (Baiet et al. 2011; also on P30 [103].
These typos were removed, line 525 page 24; line 435 page 20.
References: Why sometimes hyperlinks?
It was probably an EndNote formatting mistake. Hyperlinks were removed as suggested.
Fig3 and methods described therein: Reference is lacking. Or is this your own data? Then maybe this is something you want to publish, but not in a review?! Better ask the editor about that.
Figure 3 utilises public genomes and public software. We did not generate any new data, we just gathered and analysedpre-existing data. We think it is important to show the possible homologs of multiple microalgal species. A similar approach was used in “Mathieu-Rivet, E.; Kiefer-Meyer, M.C.; Vanier, G.; Ovide, C.; Burel, C.; Lerouge, P.; Bardor, M. Protein N-glycosylation in eukaryotic microalgae and its impact on the production of nuclear expressed biopharmaceuticals. Front Plant Sci 2014, 5, 359, doi:10.3389/fpls.2014.00359”.
Fig4: This is a huge figure with lots of details which either need to be explained or removed. E.g. B.3. sits there without any explanation. E.g. B.2.: It is not purposeful to cram CRISPR into this figure. And so on…
We added multiple references in the caption of the figure to streamline it and offer the reader possibility to look for further information. We believe that this figure is very important to explain the different successful approaches that could be used in microalgae, especially now that we shorten the descriptions in the text as recommended.
Line 558 page 26: “Protein glyco-engineering approaches (A) can target (1) the recombinant DNA sequence (rDNA) [138], (2) the sub-cellular location of the biopharmaceutical [112], and (3) the glycosylation pattern of the translated protein [139,140]. Cell glyco-engineering strategies (B) can modify the activity of glycosylation enzymes by (1) random genetic insertion [141], (2) targeted gene knock-in or knock-out [142-146], and (3) inhibitor interference [139]”.
P28/29: Contradiction: “This strategy…without changing its .. activity… The first strategy is used … to enhance their activity…” ? The activity of a protein can be affected by both, its aa composition and its glycosylation.
We changed the first “activity” with “function”.
Line 575, page 26: “This strategy is based on changing the amino acid sequence of a recombinant protein without changing its structure or function in the cell, to either (i) increase the number of glycans present or (ii) remove glycan attachment sites. The first strategy is used in the case of non-immunogenic recombinant glycoproteins in order to enhance their activity…”
P29: What is the difference between reducing and removing a glycosylation site?
We removed that part in the revised version of the manuscript to shorten long technological descriptions as suggested by reviewer 2 ; however by “reducing” we meant reducing the total number of glycosylation sites and by “removing”, we meant completely eliminate all the glycosylation sites, to obtain a non-glycosylated protein.
P32, 5.2.: Once more, here and throughout the manuscript: glycoengineering is not only about reducing immunogenicity, but e.g. also about activity (see e.g. rise in ADCC in the case of non-fucosylated Herceptin)
This comment relates to previous comments from reviewer 1 and 2. We agree with reviewer 1 and 2, therefore we added multiple parts throughout the review highlight the importance of glycosylation (and thus glyco-engineering) for activity and folding of recombinant proteins (and not only immunogenicity).
Line 245, page 12: “Glycosylation significantly enhances yield, folding, efficacy, and pharmacokinetics of recombinant biopharmaceuticals [30]. During recombinant protein production, non-human host organisms can attach glycan residues (monosaccharides) that would be absent on the human endogenous protein, potentially resulting in lower yields, incorrect folding, and inefficacy of the biopharmaceutical [10]. For example, glycosylation plays a fundamental role in activity of antibody-based therapeutics [88]. In fact, antibody are able to target and kill hostile cells by antibody-dependent cellular cytotoxicity (ADCC) or complement-dependent cytotoxicity (CDC) mechanisms, and both ADCC and CDC are directly related to glycosylation presence and status of the antibody [89-91]. Therefore, an antibody carrying incorrect glycosylation might show diminished activity.
Line 530 page 24: “Although the computational analysis suggests that several model algal species possess promising characteristics for biopharmaceutical production, including the presence of GnT I, several potentially problematic enzymes are also present. Albeit antigenicity of recombinant biopharmaceuticals produced in microalgae has been tested (mostly to assess proper folding and showing mixed results) [57,59,64], immunogenicity of microalgal-based therapeutics has never been tested. However, given the diversity of microalgal glycans and the predicted differences with human glycosylation profiles, “humanization” of microalgae glycans via glyco-engineering is likely needed to elicit proper folding and remove possible immunogenic glycans, to lastly produce active and safe recombinant biopharmaceuticals from microalgae [4]”.
Line 543 page 24: “In biofactories such as E. coli, yeasts, CHO cells, and plants, different glyco-engineering techniques have been successfully used to manipulate and “humanize” glycans to produce active and safe biopharmaceuticals for patients”.
Line 678 page 30: “The inserted DNA encodes specific enzymes absent in the wild type organism that will add the desired residues to the recombinant glycoprotein to obtain properly folded, active, and safe recombinant biopharmaceuticals”.
P33: [136] and papers cited therein does, as far as I think I understood, not describe the use of inhibitors in biopharmaceutical production in CHO but only in small scale experiments. Right? In that case, your sentence would be misleading.
We removed that part, line 658 page 29.
P37: “Both… dependent [what?] are missing a …”
We added the word “species”, line 762 page 33.

Round 2
Reviewer 2 Report
All much better - just a remark: The "yellow square" is lacking in the legend of Fig 2, maybe mend this.